# BCEPS: A Web Server to Predict Linear B Cell Epitopes with Enhanced Immunogenicity and Cross-Reactivity

**DOI:** 10.3390/cells10102744

**Published:** 2021-10-14

**Authors:** Alvaro Ras-Carmona, Hector F. Pelaez-Prestel, Esther M. Lafuente, Pedro A. Reche

**Affiliations:** Laboratory of Immunomedicine, Department of Immunology & O2, Faculty of Medicine, University Complutense of Madrid, Ave Complutense S/N, 28040 Madrid, Spain; aras@ucm.es (A.R.-C.); hpelaez@ucm.es (H.F.P.-P.); melafuente@med.ucm.es (E.M.L.)

**Keywords:** B cells, epitopes, prediction, machine learning, SARS-CoV-2

## Abstract

Prediction of linear B cell epitopes is of interest for the production of antigen-specific antibodies and the design of peptide-based vaccines. Here, we present BCEPS, a web server for predicting linear B cell epitopes tailored to select epitopes that are immunogenic and capable of inducing cross-reactive antibodies with native antigens. BCEPS implements various machine learning models trained on a dataset including 555 linearized conformational B cell epitopes that were mined from antibody–antigen protein structures. The best performing model, based on a support vector machine, reached an accuracy of 75.38% ± 5.02. In an independent dataset consisting of B cell epitopes retrieved from the Immune Epitope Database (IEDB), this model achieved an accuracy of 67.05%. In BCEPS, predicted epitopes can be ranked according to properties such as flexibility, accessibility and hydrophilicity, and with regard to immunogenicity, as judged by their predicted presentation by MHC II molecules. BCEPS also detects if predicted epitopes are located in ectodomains of membrane proteins and if they possess N-glycosylation sites hindering antibody recognition. Finally, we exemplified the use of BCEPS in the SARS-CoV-2 Spike protein, showing that it can identify B cell epitopes targeted by neutralizing antibodies.

## 1. Introduction

Adaptive immunity in vertebrates is mediated by B and T cells, which recognize antigens with exquisite specificity [1,2,3]. B cells recognize freely accessible antigens via their B cell receptor (BCR) and are responsible for the humoral immunity. The BCR comprises a membrane-bound immunoglobulin that it is secreted after B cell activation and differentiation as a soluble immunoglobulin or antibody [4,5]. In this context, a B cell epitope is the specific portion of the antigen recognized by a BCR or its derived antibody.

Protein-derived B cell epitopes can be classified in two types: conformational or linear. Conformational epitopes, also known as discontinuous epitopes, encompass residues that are not sequential in the primary structure; thus, their antigenic reactivity depends on the native conformation of the protein [6]. In contrast, linear epitopes, also named as continuous epitopes, consist of residues that are sequential [2]. However, in a linear B cell epitope, not every single residue necessarily contacts the antibody paratope [6]. It is often considered that most B cell epitopes are conformational. However, the majority of B cell epitopes reported and deposited in databases are linear and it has been postulated that they likely result from antigens degraded proteolytically by various immune cells and subsequently exocytosed [7]. The name of linear B cell epitopes can also be misleading, since they are recognized by cognate antibodies in particular three-dimensional conformations [8] that can nonetheless be reproduced by a synthetic peptide without the remaining protein context.

B cell epitope identification is of practical interest for producing antibodies with a desired specificity and it is of potential interest for vaccine design. There are diverse experimental methods to identify B cell epitopes [7,9] but they all demand much time and resources. Therefore, there is a need for developing computational methods that can facilitate their identification. The possibility of predicting linear B cell epitopes is of particular practical interest since, unlike conformational B cell epitopes, they can be recognized by antibodies isolated from the protein context as synthetic peptides [2,7,10]. B cell epitopes can be predicted using different available methods, including some based on machine learning [2,4,8,9,11,12,13,14]. In general, however, all the available methods for linear B cell epitope prediction are quite unreliable, with close to random predictions [12]. In addition, it has been noted that linear B cell epitopes do often fail to generate cross-reactive antibodies with the native antigens and are poorly immunogenic [7]. Thereby, there is a need for more accurate B cell prediction methods and tools, capable of discriminating epitopes candidates suitable for practical applications. 

Here, we present BCEPS (B Cell Epitope Prediction Software), a web server for the prediction of linear B cell epitopes within protein sequences. BCEPS relies on machine learning based models trained on linearized B cell epitopes extracted from the tertiary structures of antibody–antigen complexes. The top-performing model implemented in BCEPS was generated using Support Vector Machine and reached an accuracy on 10-fold cross validation experiments of 75.38% ± 5.02. This model outperformed related methods when tested in an independent dataset. BCEPS also facilitates the selection of B cell epitopes with crucial structural features (e.g., flexibility and accessibility) that make them more suitable to generate antibodies that are likely cross-reactive with the native antigens. Moreover, BCEPS can also detect those B cell epitopes that include motifs for presentation by major histocompatibility complex class II (MHC II) molecules and can thereby recruit help from CD4 T cells, hence being more immunogenic. BCEPS is available for free public use at http://imbio.med.ucm.es/bceps/ (accessed on 29 September 2021).

## 2. Materials and Methods

### 2.1. Acquisition of B Cell Epitopes

B cell epitopes were extracted from antigen–antibody tertiary structures, downloaded from the abYbank/AbDb database [15]. For any given antigen–antibody structure, the B cell epitope sequence consisted of all antigen residues within a range of 4 Å radius to any antibody atom in sequential order. These linearized discontinuous B cell epitopes were used for model building and optimization. For additional validations, we obtained experimentally verified B cell epitopes from the Immune Epitope Database (IEDB) [16,17], considering only those with positive assays and discarding those that were not associated to an UNIPROT accession number. Linear and discontinuous B cell epitopes were retrieved and kept separately in distinct datasets. In addition, discontinuous B cell epitopes were put into sequential amino acid sequences (linearized). Non-B cell epitopes were randomly obtained from the same antigen sequences than the corresponding B cell epitopes.

### 2.2. Sequence Similarity Reduction and Similarity Analysis

Sequence redundancy in all datasets was reduced using CD-HIT [18] so that amino acid sequence identity was <80%. Sequence similarity in datasets was analyzed after pairwise sequence alignments generated using the Needleman–Wunsch global alignment algorithm implemented in the Biopython package [19]. To obtain a measure of sequence similarity in a dataset, all sequences were aligned pairwise, but with themselves (for a dataset with N sequences, there will be N × N-1 alignments). Identities were obtained for each alignment and the average identity was computed.

### 2.3. Building and Optimization of B Cell Epitope Prediction Models

Prediction models were built and evaluated using the Waikato Environment for Knowledge Analysis (WEKA) package [20] under classification settings. WEKA provides a large collection of machine learning algorithms (MLAs) for data classification. Here, Support Vector Machine (SVM)—with both Gaussian radial basis function (RBF) Kernel and polynomial function–, Neural Network (NN), K-Nearest Neighbor (KNN,) and Random Forest (RF) were used. Input data for WEKA consisted of datasets in ARFF format with epitope sequences translated into amino acid composition, dipeptide composition, and the combination of both, as described elsewhere in Ras-Carmona et al.’s study [21]. Briefly, the amino acid composition translation produces a vector feature including the proportion of each of the 20 natural amino acids in any sequence. The dipeptide composition translation produces a vector including the proportion of all possible pairs of amino acids (20 × 20). Combining amino acid and dipeptide composition produces a feature vector with the corresponding 420 values. Diverse models were trained by varying the parameters that define the MLAs. We specifically varied the complexity parameter of SVMs along with the gamma value of the RBF Kernel and the exponent of the Polynomial function. For NN, RF, and KNN we varied the number of hidden nodes, decision trees, and neighbors, respectively. All models were evaluated in 10-fold cross-validation classification experiments that were repeated 10 times. 

### 2.4. Measures of Performance

To obtain the performance of the models, sensitivity (SE), specificity (SP), accuracy (ACC), and Matthew’s correlation coefficient (MCC) were computed using Equations (1)–(4), respectively. These measurements are expressed in terms of true positive (TP), false negative (FN), true negative (TN), and false positive (FP) predictions resulting from classification experiments.
(1)$$\mathrm{SE}=\frac{\mathrm{TP}}{\mathrm{TP}+\mathrm{FN}}$$
(2)$$\mathrm{SP}=\frac{\mathrm{TN}}{\mathrm{TN}+\mathrm{FP}}$$
(3)$$\mathrm{ACC}=\frac{\left(\mathrm{TP}+\mathrm{TN}\right)}{\left(\mathrm{TP}+\mathrm{FP}+\mathrm{TN}+\mathrm{FN}\right)}\;\times \;100$$
(4)$$\mathrm{MCC}=\frac{\left(\mathrm{TP}\;\times \;\mathrm{TN}\right)-\left(\mathrm{FN}\;\times \;\mathrm{FN}\right)}{\sqrt{\left(\mathrm{TN}+\mathrm{FN}\right)\left(\mathrm{TP}+\mathrm{FN}\right)\left(\mathrm{TN}+\mathrm{FP}\right)\left(\mathrm{TP}+\mathrm{FP}\right)}}$$

### 2.5. Prediction of Linear B Cell Epitopes with Freely Available Tools

For comparative analyses, linear B cell epitopes were predicted using BepiPred [22,23], IBCE-EL [24], and LBtope [25]. BepiPred predictions were performed using a standalone version of BepiPred (Bepipred 2.0). For any peptide sequence, BepiPred assigns B cell epitope propensities per residue and average B cell epitope scores were computed. Peptide sequences with B cell epitope scores ≥ 0.5 were considered B cell epitopes. B cell epitope predictions with IBCE-EL and LBtope were carried out online at the web sites http://www.thegleelab.org/iBCE-EL/ (accessed on 4 May 2021) and https://webs.iiitd.edu.in/raghava/lbtope/peptide.php (accessed on 4 May 2021), respectively. LBtope predictions were obtained under the mode for multiple peptide submissions, using the default model labeled as “LBtope_Variable (original dataset)”. Peptides with a percent of probability higher than 0.6 were considered B cell epitopes, as predefined in the web. In IBCE-EL, multiple peptide sequences were submitted in FASTA format and peptides indicated in the IBCE-EL result page as “BCE” were considered B cell epitopes. 

### 2.6. Prediction of Peptide Immunogenicity and Computation of Population Protection Coverage 

B cell epitope immunogenicity is contingent on their capacity to recruit CD4 T helper (Th) cells, which requires binding and presentation by major histocompatibility complex class II (MHC II) molecules. In BCEPS, we enabled peptide binding predictions using local standalone versions of RANKPEP [26,27] and NetMHCIIpan [28] to 8 mouse MHC II molecules (H-2-IAb, H-2-IAd, H-2-IAk, H-2-IAq, H-2-IAs, H-2-IAu, H-2-IEd, and H-2-IEk) and 35 common human MHC II molecules encompassing HLA-DRB1 chains. The HLA-DRB1 molecules targeted for binding were the following: DRB1*13:01, DRB1:04:01, DRB1*12:01, DRB1*08:01, DRB1*04:02, DRB1*11:02, DRB1*01:01, DRB1*12:02, DRB1*16:01, DRB1*01:03, DRB1*15:02, DRB1*16:02, DRB1*08:02, DRB1*13:02, DRB1*07:01, DRB1*04:04, DRB1*03:01, DRB1*15:03, DRB1*11:03, DRB1*09:01, DRB1*03:02, DRB1*11:01, DRB1*13:03, DRB1*11:04, DRB1*04:08, DRB1*08:04, DRB1*04:03, DRB1*08:03, DRB1*10:01, DRB1*01:02, DRB1*04:07, DRB1*04:05, DRB1*14:01, DRB1*14:02, and DRB1*15:01. A peptide is considered to bind to a particular MHC II molecule if it reaches a relative percentile rank above 10% with either of the two tools. Peptides binding to MHC II molecules have a variable size (between 9 and 22 residues), but include a 9-mer residue core which sits on the MHC II molecule [29]. HLA II binding predictions were carried out to report 15-mer peptides and any predicted B cell epitope encompassing the corresponding 9-mer residue was considered to bind to the relevant HLA II molecule. 

In humans, the population protection coverage (PPC) of a B cell epitope is defined as the proportion of the population in which that epitope could be immunogenic—elicit specific antibodies—as judged by their ability to recruit help from Th cells. Hence, in BCEPS, the PPC of the epitopes is computed after their HLA-DRB1 binding profiles as indicated elsewhere in [30,31] for 4 distinct ethnic groups in North America (Caucasian, Afroamerican, Asian, and Native North Americans), reporting an average value. Genetic frequencies of HLA-DRB1 alleles required for PPC calculations were obtained from http://www.allelefrequencies.net (accessed on 25 May 2016) and are provided in the Appendix A. 

### 2.7. Ectodomain Location and Prediction of Flexibility, Accessibility, Hydrophilicity, and Glycosylation Sites

In BCEPS, input amino acid sequences were subjected to predictions to detect peptide leader sequences, transmembrane helical regions, and Glycosylphosphatidylinositol (GPI) anchoring regions, using SIGNALP [32], TMHMM [33], and big-Pi [34]. After these predictions, BCEPS identifies as ectodomain residues those located in the mature portion of secreted proteins and/or in solvent exposed regions of membrane bound proteins. 

Likewise, relative solvent accessibility (RSA) and normalized B values—used as a measure of flexibility—per residue were predicted for the entire input amino acid sequence using NetSurfP [35] and profBval [36], respectively. The Hopp and Woods scale [37] was used to assign hydrophilicity to amino acid residues. Subsequently, BCPES obtained measures of epitope flexibility, accessibility, and hydrophilicity consisting of average values computed from the corresponding epitope residue values. N-glycosylation sites in input amino acid sequences and the corresponding epitopes were predicted using NetNGlyc [38].

## 3. Results

### 3.1. Strategy to Generate B Cell Epitope Prediction Models

We approached the task of developing B cell epitope prediction models as a classification problem for machine learning (ML). Under this approach, B cell epitope prediction models were generated by training ML algorithms (MLAs) to distinguish B cell epitopes from non-B cell epitopes. To that end, we constructed a non-redundant dataset including the sequences of 555 B cell epitopes and 555 non-B cell epitopes. Hereafter, we will refer to this dataset as BCETD_555_. B cell epitope sequences in BCETD_555_ were obtained after the tertiary structure of antigen–antibody complexes and consisted of linearized conformational B cell epitopes, as they encompass antigen residues in contact with cognate antibodies (≤4.0 Å) ordered sequentially. All B cell epitope sequences included in the training dataset ranged from 11 to 25 residues, with a mean and median length of 16.01 ± 3.64 and 16, respectively. Sequence redundancy was avoided using CD-HIT [18], removing epitope sequences with more than 80% identity. The average sequence identity between B cell epitopes included in BCETD_555_ was 15.84 ± 5.18%. Non-B cell epitope sequences in BCETD_555_ were extracted randomly from the same antigens than B cell epitopes and they had the same size distribution as B cell epitope sequences (min: 11, max: 25, mean: 16.01 ± 3.64 and median: 16). The average sequence identity between non-B cell epitope sequences was 15.89 ± 4.74%. Overall, considering both B cell epitopes and non-B cell epitopes, the average sequence identity in BCETD_555_ was 15.63 ± 4.86%. BCETD_555_ is available as Appendix A at the journal website (Appendix A).

### 3.2. B Cell Epitope Prediction Models

We built and optimized B cell epitope models by training MLAs on the BCETD_555_ dataset, using features consisting of the amino acid composition, dipeptide composition, and the combination of both. We performed multiple classification experiments under 10-fold cross-validation experiments (repeated 10 times) varying the relevant parameters of the selected MLA. We specifically applied Support Vector Machine (SVM) with Gaussian radial basis function (RBF) Kernel and Polynomial function, Neural Networks (NN), Random Forest (RF), and K-Nearest Neighbors (KNN). All these MLAs are frequently applied to biological data classification and model building [39]. Judging by the ACC of the classifications, the best models were obtained by training MLAs on amino acid composition and the combination of amino acid composition and dipeptide composition (Figure 1). However, MLAs trained on dipeptide composition alone reached much modest values of ACC, highlighting the major contribution of the amino acid composition to the performance of the models. Therefore, we selected the top performing ML models trained on amino acid composition to obtain further measures of performance such as SE, SP, and MCC (Table 1). 

In general, all the MLAs analyzed had a similar performance, reaching ACC values in cross-validation above 72% (Table 1). The highest ACC was reached by an RBF SVM model, which was significantly better than that reached by KNN (*p*-value = 0.0001) and NN (*p*-value = 0.036), as judged by two sample T-tests. The ACC of the SVM model was also better than that reached by the top-performing RF model, but without statistical significance (*p*-value: 0.56). The MCC value reached by the SVM model was 0.51, which was also higher than that reached by all the other ML models. Thereby, we selected the RBF SVM model for further evaluation in independent test datasets

We also evaluated the top-performing SVM model generated here in two independent test datasets and compared the predictions with those produced by other B cell prediction tools, including BepiPred [22,23], IBCE-EL [24], and LBtope [25]. These tools are also aimed to predict linear B cell epitopes and are based on RF, SVM, and a combination of SVM and KNN, respectively. The two independent datasets were generated after experimental B cell epitope sequences retrieved form the IEDB database [16,17] (details in Material and Methods). One of the datasets included the sequences of 2195 linear B cell epitopes (IEDB Linear Epitope dataset, ILED_2195_) while the other included the sequences corresponding to 1246 discontinuous B cell epitopes; residues ordered sequentially (IEDB Discontinuous Epitope Dataset, IDED_1246_). Each dataset included an equal number of non-B cell epitope sequences that were obtained randomly from the same antigens and with the same size than the counterpart B cell epitopes. We only considered B cell epitopes with more than 10 residues and less than 26, to resemble the size of the epitopes in the BCETD_555_ dataset. The average sequence identity in the ILED_2195_ and IDED_1246_ datasets was 15.31 ± 5.06% and 15.52 ± 4.78%, respectively. The average sequence identity considering just B cell epitopes was 15.31 ± 5.47% and 15.85 ± 4.99% in the ILED_2195_ and IDED_1246_ datasets, respectively. ILED_2195_ and IDED_1246_ are available as Appendix A at the journal website (Appendix A, respectively). 

The results of classifying B cell epitopes in the described independent datasets using BepiPred, LBtope, and IBCE-EL and our SVM model are summarized in Table 2. Our SVM model performed better in the IDED_1246_ dataset than in ILED_2195_, reaching ACC values of 67.05% and 60.38%, respectively, and MCC values of 0.34 and 0.21, respectively. This result is expected, since the IDED_1246_ dataset consists of sequential (linearized) conformational epitopes as it does the BCETD_555_ training dataset. The performance of the SVM model in both independent datasets was worse than under cross-validation, but yet much better than that of BepiPred, LBtope, and IBCE-EL, which were clearly unable to distinguish B cell epitopes from non-B cell epitopes in any of the two test datasets. 

### 3.3. BCEPS Web Server

We have developed a web-based tool named BCEPS for B Cell Epitope Prediction Software, which enables the prediction of B cell epitopes using the ML models developed here. BCEPS is available for free public use at http://imbio.med.ucm.es/bceps/ (accessed on 29 September 2021). BCEPS interface, shown in Figure 2A, has been designed for an easy and intuitive use. The input data for BCEPS is a protein sequence in FASTA format and B cell epitopes can be predicted using the top-performing SVM, RF, and NN models described in Table 1. BCEPS predictions are carried out on all the different peptides, with a size selected by the user, included in the input sequence. By default, BCEPS returns all peptides indicating whether they are predicted to be B cell epitopes (B cell epitope score ≥ threshold). However, BCEPS can only show predicted B cell epitopes if that option is selected. Moreover, since various consecutives peptides can be B cell epitopes, BCEPS has two options to simplify the output: “Extended B cell epitopes” and “Collapsed B cell Epitopes”. If the option “Extended B cell epitopes” is checked, consecutive peptides that are predicted as B cell epitopes are extended and reported as a single epitope. If the option “Collapsed B cell Epitopes” is selected, BCEPS assigns a B cell epitope score per residue, which is computed as the mean of the B cell epitope scores of all peptides including that residue. Then, sequential residues predicted to be B cell epitopes (threshold ≥ 0.5) are joined and BCEPS returns those ones with length ≥ than the selected size. BCEPS can also report various epitope features/properties selected by the user. These features include hydrophilicity, flexibility, accessibility, ectodomain location, glycosylation sites, and immunogenicity. Peptide hydrophilicity and flexibility are reported as numeric values computed as detailed in Material and Methods. Ectodomain location and glycosylation sites are reported as binary tags (Y/N) that serve to assess if a particular peptide is located in the solvent accessible region of a membrane-bound or secreted protein and/or if it has N-glycosylation sites, respectively (details in Material and Methods). Peptide immunogenicity is contingent on getting help from CD4 T helper (Th) cells for antibody production, which requires presentation by major histocompatibility complex class II (MHC II) molecules. Thereby, BCEPS can report peptide-MHC II binding profiles and for human MHC II molecules their combined phenotypic frequency in the population (details in Material and Methods). This value represent the proportion of the population in which the peptide will be immunogenic in humans (population protection coverage). In the output (Figure 2B), all the selected features are listed per peptide in an interactive table, allowing users to sort and/or filter the predicted B cell epitopes. 

### 3.4. Case Study: SARS-CoV-2 Surface Spike Glycoprotein

Severe acute respiratory syndrome coronavirus 2 (SARS-CoV-2) virus is the cause of the ongoing COVID-19 pandemic. SARS-CoV-2 infects host cells that express the receptor angiotensin-converting enzyme 2 (ACE2), which is engaged by SARS-CoV-2 Spike (S) protein [40]. Blocking the interaction between SARS-CoV-2 and ACE2 prevents viral entry [41,42,43]. Hence, current COVID-19 vaccines aim to generate neutralizing antibodies using the S protein [44,45,46]. Nonetheless, it is also now clear that these vaccines induce considerable CD8 T cell responses which also ought to contribute to the reported protection against COVID-19 [47,48,49]. SARS-CoV-2 S protein is highly glycosylated, which hinders antibody recognition. Therefore, neutralizing antibodies have been shown to target non-glycosylated epitopes located mainly in the Receptor Binding Domain (RBD) of the S protein [50]. A number of such neutralizing B cell epitopes have been deposited and are available at IEDB (34). Here, we examined the utility of BCEPS for identifying known neutralizing B cell epitopes in the SARS-CoV-2 S protein.

We entered the entire amino acid sequence of SARS-CoV-2 S protein sequence (ACN: YP_009724390.1) into BCEPS and predicted B cell epitopes selecting the length of 18 residues and the option to extend B cell epitopes. As a result, we obtained a set of 38 predicted B cell epitopes and of those, 21 are located in the ectodomain and do not contain any N-glycosylation site (Appendix A). In BCEPS, potential B cell epitopes can be ordered by solvent accessibility and flexibility and we selected for further investigation 10 of them, exhibiting an accessibility above 0.20 or a flexibility higher than 0.74. Subsequently, we searched the IEDB database [16,17] for B cell epitopes from SARS-CoV-2 using the amino acid sequence of the 10 selected peptides, finding that 6 of them had coincidences (≥8 identical residues) with B cell epitopes known to be targeted by neutralizing antibodies (Table 3). 

We also inspected the location of predicted B cell epitopes matching known neutralizing epitopes in the tertiary (3D) structure of SARS-CoV-2 S protein (Figure 3). As indicated earlier, three of these epitopes (NNLDSKVGGNYNYLYR, FRKSNLKPFERDISTEIYQA, and TEIYQAGST) map in the RBD region of SARS-CoV-2 S protein, highlighting their neutralizing nature. Peptide ALHRSYLTPGDSSSG maps in a region proximal to the RBD domain and it is expected that antibodies recognizing this epitope will prevent binding to ACE2 by steric hindrance and hence be neutralizing as well. In contrast, the epitopes QTQTNSPRRARSVAS and PAQEKNFTTAP are located far from the RBD domain and their neutralizing nature is less evident. However, we can speculate that antibodies recognizing these epitopes may interfere with conformational changes that are required to expose the RBD domain and engage ACE2 during viral entry [51]. It is important to note that the neutralizing B cell epitopes that are predicted by BCEPS are clearly exposed. Moreover, all of them, except for PAQEKNFTTAP, which is partially located in a beta sheet, lie completely in long flexible loops. Therefore, we could expect that synthetic peptides corresponding to these B cell epitopes could elicit antibodies that are cross-reactive with the native antigen.

## 4. Discussion

Producing entire antigens for antibody production is not a trivial matter. Thereby, the main practical objective of B cell epitope prediction is to identify a portion of the antigen that can substitute it in the production of specific antibodies. Conformational or discontinuous B cell epitopes are arguably the most relevant for antibody/B cell recognition but, unfortunately, isolating conformational B-cell epitopes from their protein context is a difficult task that requires suitable scaffolds for epitope residue grafting [2]. As a result, prediction of conformational B-cell epitopes has currently had little practical translation in the antibody-production industry. In contrast, linear B cell epitopes, can be isolated from their protein context and used (e.g., as synthetic peptides) to replace the counterpart antigens for antibody production and detection. Therefore, there is a practical interest in developing methods to predict linear B cell epitopes and here, we have introduced a new web-based tool, BCEPS, for such a task (available for free public use at http://imbio.med.ucm.es/bceps/, accessed on 29 September 2021). 

BCEPS relies in ML-based models that were trained on linearized conformational B cell epitopes obtained from available 3D structures of antigen–antibody complexes. The top-performing model implemented in BCEPS, based on SVM, reached an accuracy under cross-validation of 75.38% ± 5.02. In an independent test set, consisting of B cell epitopes obtained from IEDB, the accuracy of this model dropped to 67.05%, but yet, clearly outperformed related tools such as BepiPred [22,23], IBCE-EL [24], and LBtope [25], which were unable to distinguish B cell epitopes from non B cell epitopes. Therefore, the predictive power of BCEPS is notable. Moreover, it is unlikely that B cell epitope prediction methods can be more accurate since the BCR/antibody repertoire is so diverse that almost anything can be recognized; there are no, or few, true non-B cell epitopes.

In practice, the extreme diversity of the BCR repertoire determines that specific antibodies can be produced against any 10–15 residue-long peptide [52]. As a result, the selection of B cell epitopes that are cross-reactive with the native antigen become of paramount relevance. Thereby, we have enhanced BCEPS with features allowing such selection. Cross-reactive linear B cell epitopes must be solvent accessible and located in flexible regions, and thus BCEPS permits sorting/selecting peptides according to these properties. Membrane-bound proteins and secreted proteins are often glycosylated, which also hinder cross-reactive recognition by antibodies elicited by synthetic peptides. Consequently, BCEPS allows discarding peptides with predicted N-glycosylation sites in the ectodomain of proteins. By applying these selection criteria, we have shown that it is possible to identify known linear B cell epitopes in SARS-CoV-2 S protein, which are targeted by neutralizing antibodies. As expected, these B cell epitopes are located in solvent exposed flexible loops of SARS-CoV-2 S protein (Figure 3). It is worth noting that SARS-CoV-2 Spike protein also exhibits many other neutralizing conformational B cell epitopes (about 150 of them with ≥10 residues can be identified in the IEDB resource) that are out of the reach of BCEPS. Whether these conformational epitopes are of greater functionally/relevance than the linear B cell epitopes is an open question that will require experimental scrutiny. However, unlike linear B cell epitopes, these conformational B cell epitopes in SARS-CoV-2 Spike protein cannot be used to produce antibodies alone, unless they are engineered in adequate scaffolds retaining the epitope 3D-structure. Likewise, it also worth remarking that BCEPS, or any B cell epitope prediction method, bypass the need for experimental verification. Therefore, B cell epitopes predicted by BCEPS within a given query antigen will need to be synthesized and antibodies produced against them in animal models will need to be tested for their ability to recognize the antigen. These B cell epitope may be useful for epitope vaccine only if the raised antibodies have some relevant biological activity against the corresponding pathogen, e.g., blocking viral entry. Currently, there is no commercial vaccine available based on B cell epitope predictions. Moreover, depending on the infection/pathogen, inducing antibodies with the desired specificity can be an important step towards developing a vaccine, but antibodies alone may not suffice.

Although peptides can be antigenic, they exhibit little immunogenicity. They can be recognized by B cells, but such recognition is not enough to elicit the production of antibodies. B cells require stimulation from T helper (Th) cells to produce antigen-specific antibodies. To that end, B cells must present to Th cells peptides derived from BCR-recognized antigens bound to MHC II molecules. Antigen/peptide immunogenicity is thus concomitant with antigen-presentation by MHC II molecules. Since MHC II molecules can only bind peptides containing specific motifs, the chance that these motifs to realize in short peptides is small, which explains their poor immunogenicity. The immunogenicity of peptides can be enhanced by various means, including by fusion with larger proteins. However, some peptides can themselves bind to MHC II molecules, which will make them inherently more immunogenic. Thereby, in BCEPS, we enabled the identification of B cell epitopes that can potentially bind and be presented by MHC II molecules, in humans known as Human Leukocyte Antigen class II (HLA II) molecules. MHC II molecules encompass two chains, alpha and beta, both contributing to delineate the peptide binding groove. In humans, HLA II molecules are polygenic, including HLA-DP, HLA-DQ, and HLA-DR molecules. Moreover, they are highly polymorphic and the polymorphisms determine their peptide binding specificity [53]. In BCEPS, we chose to predict peptide binding to HLA-DR molecules with the beta chain encoded by HLA-DRB1 gene alleles for two reasons: the alpha chain is non polymorphic [53] and HLA-DRB1 expression in the cell surface of antigen presenting cells is higher than that of other HLA II molecules [54]. Another relevant feature of HLA II molecules is that distinct allelic variants are expressed in the population with uneven and variable frequencies depending on the ethnic group [31]. Since HLA II presentation determines the inherent immunogenicity of B cell epitopes, BCEPS also reports the percentage of the population in which a B cell epitope will alone (e.g., as a synthetic peptide) induce antibody production. This value is computed after the genetic frequently of the relevant HLA-DRB1 alleles in four distinct ethnic groups in the USA (Caucasian, Afro-American, Asians, and Native Americans) (details in Material and Methods). 

To make the most of BCEPS predictions, the output is interactive and users can filter and sort B cell epitopes according to the various features described above. Altogether, BCEPS ought to become a reference tool for B cell epitope prediction and selection.

## Figures and Tables

**Figure 1 cells-10-02744-f001:**
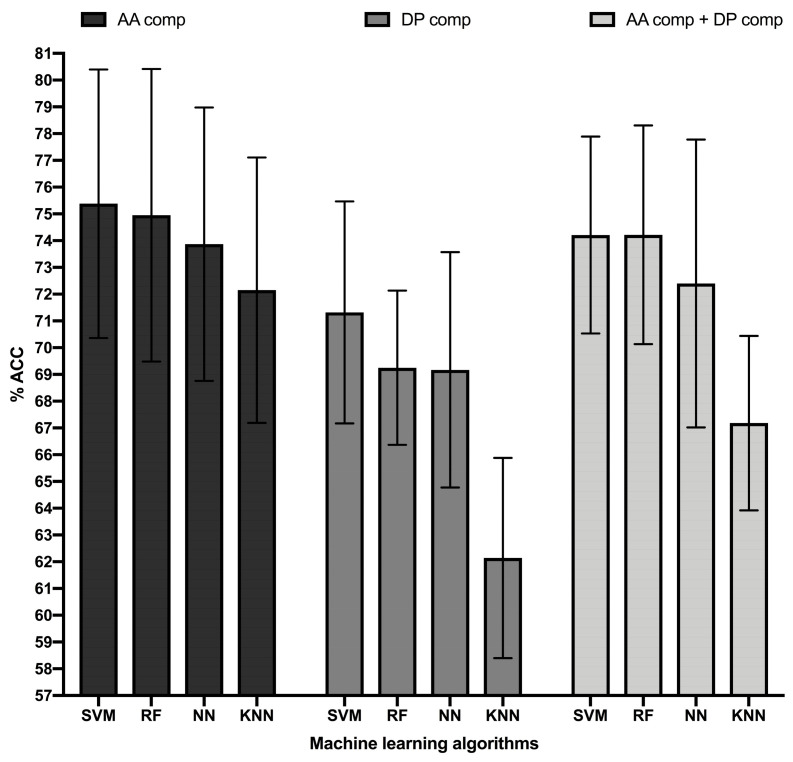
Accuracy top-performing ML models classifying B cell epitopes. Figure shows the accuracy in percentage (% ACC, *Y*-axis) of the top-performing ML models (*X*-axis) trained in B cell epitope sequence features consisting of amino acid (AA) composition (black bars), dipeptide (DP) composition (dark grey bars), and combination of amino acid and dipeptide composition (light grey bars). ML models consisted of Random Forest (RF), Support Vector Machine (SVM), K-Nearest Neighbor (KNN), and Neural Network (NN) obtained with the following parameters: RF, 450 single decision trees; SVM, RBF Kernel with a complexity parameter of 4.0 and a gamma value of 0.4, KNN, 10 neighbors; NN, Multilayer Perceptron with a single hidden layer of 11 nodes and a learning rate of 0.01. Accuracy was computed in 10-cross validation experiments repeated 10 times and error bars represent standard deviations.

**Figure 2 cells-10-02744-f002:**
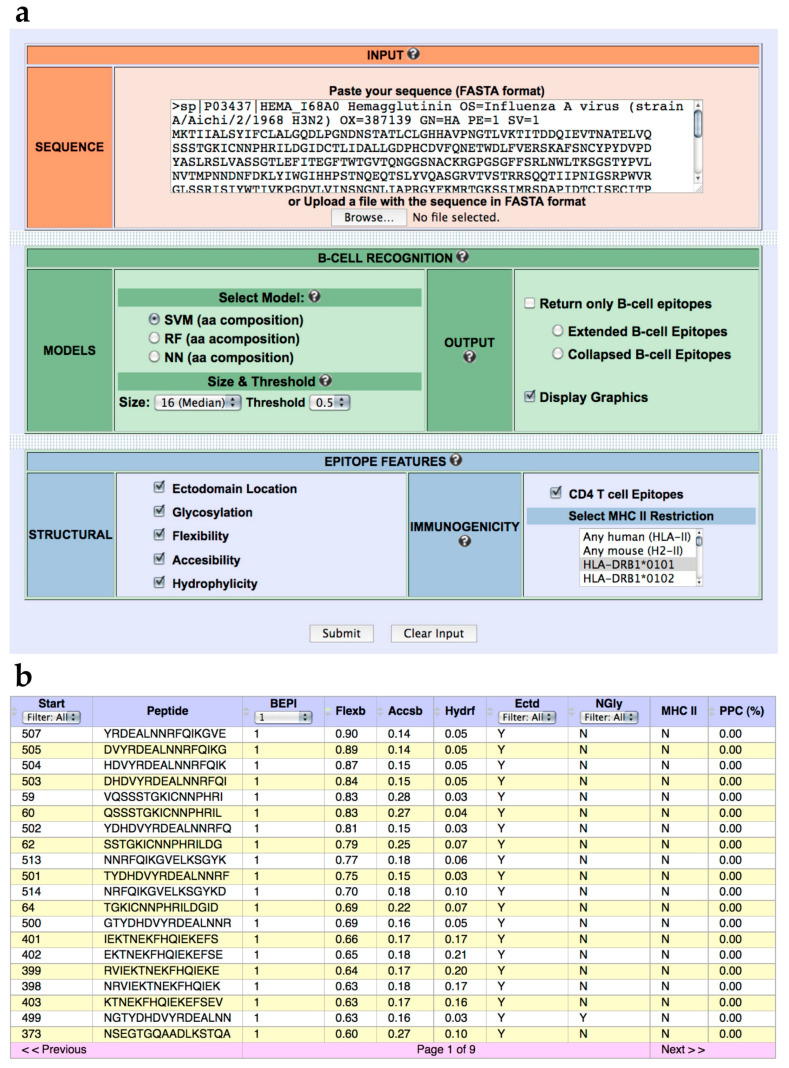
BCEPS web server. (**a**) BCEPS interface. (**b**) Representative BCEPS output obtained with default settings and the option “Extended B cell epitopes”. BCEPS main result consists of a table displaying the following information (from left to right): Peptide starting position; peptide sequence; B cell epitope prediction (1/0); flexibility value; accessibility value; hydrophilicity value; ectodomain location (Y/N); presence of N-glycosylation sites (Y/N); user selected MHC II molecules predicted to bind the peptide; and population protection coverage (only applies to human MHC II molecules).

**Figure 3 cells-10-02744-f003:**
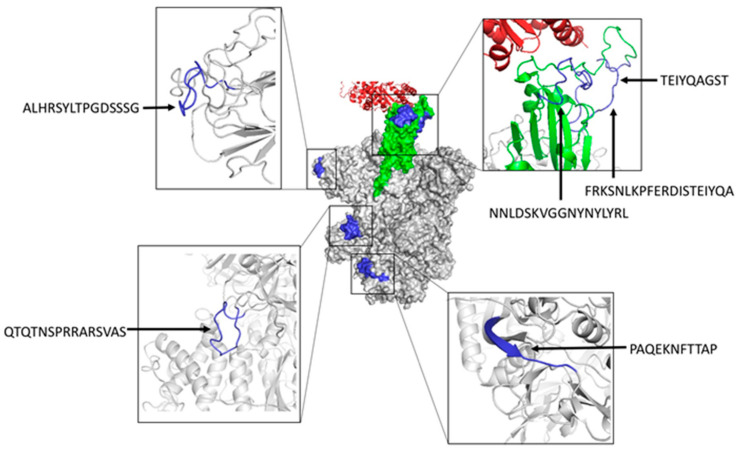
Structural mapping of neutralizing B cell epitopes in SARS-CoV-2 S protein predicted with BCEPS. The center of the figure depicts the molecular surface of the trimeric S protein in the open state (PDB: 6ACJ) with the RBD domain shown in green engaging ACE2 (shown in red ribbons). Predicted B cell epitopes overlapping with known neutralizing B cell epitopes are shown in blue. B cell epitopes are zoomed in and shown in ribbon rendering (colored in blue) around the central figure. Molecular renderings were generated using the PyMol Molecular Graphics System, Version 1.2r3pre, Schrödinger, LLC 2015 (New York, NY, USA).

**Table 1 cells-10-02744-t001:** Performance of selected ML models trained in amino acids composition.

MLAs	SE	SP	MCC	% ACC
SVM	0.73 ± 0.06	0.78 ± 0.06	0.51 ± 0.10	75.38 ± 5.02
RF	0.74 ± 0.06	0.75 ± 0.07	0.50 ± 0.11	74.95 ± 5.47
NN	0.71 ± 0.07	0.71 ± 0.08	0.48 ± 0.10	73.87 ± 5.11 ^*^
KNN	0.65 ± 0.06	0.79 ± 0.07	0.45 ± 0.10	72.15 ± 4.96 ^**^

Table reports the sensitivity (SE), specificity (SP), accuracy (ACC), and Matthew’s correlation coefficient (MMC) of top performing ML models trained and optimized in amino acid composition. Values of performance were obtained with the following parameters: SVM, RBF Kernel with a complexity parameter of 4.0 and a gamma value of 0.4; RF, 450 single decision trees; NN, Multilayer Perceptron with a single hidden layer of 11 nodes and a learning rate of 0.01; KNN, 10 neighbors; Performance values were obtained under 10-cross validation experiments repeated 10 times. ACC values labeled with asterisks are significantly smaller than those achieved by the SVM model, as judged by two sample T-tests (* *p*-value < 0.05, ** *p*-value < 0.01).

**Table 2 cells-10-02744-t002:** Comparitive performance of B cell epitope prediction methods.

Independent Test Dataset	Model/Tool	SE	SP	% ACC	MCC
ILED_2195_	SVM model	0.50	0.71	60.38	0.21
	BepiPred	0.24	0.43	33.14	−0.34
	LBtope	0.36	0.58	47.08	−0.06
	IBCE-EL	0.64	0.33	48.15	−0.04
IDED_1246_	SVM model	0.63	0.71	67.05	0.34
	BepiPred	0.42	0.52	48.11	−0.04
	LBtope	0.40	0.74	56.80	0.14
	IBCE-EL	0.86	0.20	53.20	0.09

Table reports the sensitivity (SE), specificity (SP), accuracy (% ACC), and Matthew’s correlation coefficient (MMC) reached in the independent ILED_2195_ and IDED_1246_ datasets by the our top-performing SVM model and BepiPred, LBtope, and IBCE-EL. B cell epitope predictions with BepiPred, LBtope, and IBCE-EL were carried out at the tool web sites (details in Materials and Methods).

**Table 3 cells-10-02744-t003:** B cell epitopes predicted by BCEPS matching known neutralizing B cell epitopes.

Pos	Predicted B Cell Epitope	Flex	Access	Matching IEDB Epitopes Recognized by Neutralizing Antibodies	IEDB ID
243	**ALHRSYLTPGDSSSG**WTAGAAAYY	−0.16	0.25	**ALHRSYLTPGDSSSG**	1334452
433	VIAWNS**NNLDSKVGGNYNYLYRL**	0.12	0.21	**NNLDSKVGGNYNYLYR**	1334470
456	**FRKSNLKPFERDISTEIYQA**	0.99	0.19	L**FRKSNLKPFERDIS**	1334467
				**DISTEIYQA**GSTPCNGVEGFNCYFPLQSYGFQPTNGVGYQPYRVVVL	1336532
462	KPFERDIS**TEIYQAGSTP**	0.91	0.21	**TEIYQAGST**	1335256
666	IGAGICASY**QTQTNSPRRARSVAS**QSIIAYT	0.06	0.27	**QTQTNSPRRARSVAS**	1334479
1069	**PAQEKNFTTAP**AICHDGK	−0.03	0.21	VTYV**PAQEKNFTTAP**	1313930

Table reports the position, sequence, flexibility (Flex), and solvent accessibility (Access) of predicted B cell epitopes from SARS-CoV-2 S protein that coincide with known neutralizing B cell epitopes in at least 8 residues. IEDB epitope accession numbers (IEDB ID) of neutralizing B cell epitopes are indicated. Regions overlapping between predicted and known neutralizing B cell epitopes are in bold.

## Data Availability

All data are available online as Appendix A.

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
