# Peer review of "BCEPS: A Web Server to Predict Linear B Cell Epitopes with Enhanced Immunogenicity and Cross-Reactivity"

_cells, 2021, doi:10.3390/cells10102744_

Round 1
Reviewer 1 Report
This paper by Ras-Carmona and collogues describes a new database to predict epitopes for B cell immunoglobulins. This is a very interesting tool to identify specific structures with in Igs.
This is not really a research paper rather than a resource paper.
Therefore there is not much to complain.
Author Response
Reviewer #1 has no concerns and we thank him/her for assessing our manuscript and for his/her words of appreciation.
Reviewer 2 Report
This is an interesting paper, focusing mainly on the prediction of B cell epitopes, lined to a web tool made available by the authors. Interesting but not unique as the theoretic analysis of B cell epitopes is a widely studied field now. Linking the topic to an easy-accessible web tool however increases the significance of the paper.
One weakness for the use of such approach for real-life immunology is the limitation to ‘linear’ epitopes. It’s correctly outlined by the authors’ hat even these linear epitopes can have a conformation, but the approach does exclude a lot of immunological relevant epitopes crucial in vaccine approaches.
The second weakness of the paper, shared with other similar papers is the fact that the authors mention in the introduction that B cell epitope prediction is important for many clinical approaches such as vaccine development. In order to make this convincing, the authors must provide at least one strong example of a commercial vaccine that has used B-cell prediction as a starting point to come to a final working product.
Minor question: What is the rational for working with epitopes between 10 and 26 residues?
Issue with the result section and discussion on SARS-Cov-2: the success of the mRNA approach is most likely due to the induction of MHC-I driven CD8 cytotoxic T cell activation, capable of killing infected cells, without ignoring the possible role for neutralizing Abs that would be the result of a process of protein translation and subsequent antigen presentation and MHC-II involvement. However, taking into account the route of administration of the currently approved vaccines, there is no reason to ignore the MHC-I involvement and there is no reason to attribute all the function to antibodies (in that case, an inactivated virus should work as well as a mRNA virus, and there is an agreement that this is not the case – for simple basic immunological reasons). Even in B cell paper, the authors should be more ‘objective/inclusive/neutral’ about this.
Also in the context of SARS-Cov-2: the finding that some of the predicted linear epitopes are confirmed targets of neutralizing antibodies (also mentioned in the discussion), does not prove that these antibodies are the once that do the major job in vivo. It could well be that there is a whole range of conformational epitopes that are also targeted by neutralizing antibodies that have a much higher functionality. How can the authors exclude this? This should be part of the discussion.
The overall ‘feeling’ after the review of this paper is that ‘Yes’ it works in one direction: if you have good neutralizing antibodies and you know the sequence and structure of the antigen, you can confirm/predict with the web tool the presence of a linear epitope. But where the article falls short (and to this reviewer this should be a serious open-minded part of the discussion) is: if one takes an antigen and predicts the linear epitope, is there any proof to shows that such epitope(s) can be used in a vaccine approach delivering the efficacy of the full protein? The theoretic discussion on the T cell epitope at the end of the discussion does not answer that question, and without that open discussion, there is a very big question mark as to how useful this approach (web tool) is for the design of the SARS-Cov-3 vaccine we are going to be needing 5 years from now. To this reviewer, this discussion needs to be a big part of the paper.
Author Response
We wish to thank reviewer #2 for his/her time and work reviewing our manuscript. We appreciate his insightful comments and thank the opportunity to address his/her concerns.
Point 1. Reviewer #2 perceives as a weakness that we focused in linear B cell epitopes.
“One weakness for the use of such approach for real-life immunology is the limitation to ‘linear’ epitopes. It’s correctly outlined by the authors’ that even these linear epitopes can have a conformation, but the approach does exclude a lot of immunological relevant epitopes crucial in vaccine approaches”
Sequential or continuous B cell epitopes are widely referred as linear B cell epitope for the entire information is within the primary (“linear”) structure and can be synthesize as peptides. However, we realized that the concept is somewhat misleading as one could wrongly infer that they have a linear conformation. Thereby, we indicated in the manuscript that linear B cell epitopes are recognized by cognate antibodies in specific three-dimensional conformations. To avoid any confusion, in the revised version of the manuscript we have added that this conformation can nonetheless be reproduced by a synthetic peptide without the remaining protein context (highlighted in green, page 2, lines 50-51).
We also concur with the reviewer that conformational or discontinuous B cell epitopes are clearly relevant for antibody and B cell recognition. However, we did not focus on the prediction of conformational B cell epitopes for practical reasons. Producing entire antigens for antibody production is not a trivial matter. Thereby, the main practical objective of B cell prediction is to identify a portion of the antigen that can substitute it in the production of specific antibodies. This is simple for linear/continuous B cell epitopes, as they can be synthesized isolated from their protein as peptides. However, isolating conformational B-cell epitopes from their protein context for selective antibody production is a difficult task that requires suitable scaffolds for epitope residue grafting. As a result, prediction of conformational B-cell prediction has currently had little practical presence in the antibody-production industry. These comments are now included in the revised version of the manuscript, under Discussion (page 11, lines 442-449, highlighted in green).
In sum, while it is true that prediction of only linear B cell epitopes may be seen as a weakness it is clearly of practical relevance. We also acknowledge that many antibodies generated against linear B cell epitopes (peptides) do not recognize the native antigens. These antibodies are still useful for some experimental techniques such as Western blots, but they have no use for other techniques requiring the recognition of native antigens (e.g. immunoprecipitations) or vaccine design. Therefore, we tailored BCEPS to identify linear B cell epitopes with enhanced chances to produce antibodies cross-reactive with the native antigens.
Point 2. Reviewer #2 perceives as a weakness that we indicated that B cell epitope prediction is important for many clinical approaches.
“The second weakness of the paper, shared with other similar papers is the fact that the authors mention in the introduction that B cell epitope prediction is important for many clinical approaches such as vaccine development. In order to make this convincing, the authors must provide at least one strong example of a commercial vaccine that has used B-cell prediction as a starting point to come to a final working product.”
We cannot but apology for overstating the relevance of B cell prediction in vaccine development. Subsequently, we revised our manuscript removing any perceived overstatement. Specifically, we made the following modifications (highlighted in green in the manuscript):
- In the abstract, we remove the word great
- In page 2 (lines 52-53), we changed the sentence “Many biotechnological and clinical strategies, such as vaccine development, share a need for epitope discovery “ to “B cell epitope identification is of practical interest for producing antibodies with a desired specificity and it is of potential interest for vaccine design”
Currently, there is no commercial vaccine available based on B cell epitopes, nor on T cell epitopes, as far as we are aware. We found several T cell epitope-based vaccines being tested in clinical trials, particularly for cancer, but none consisting of B cell epitopes. The major handicap for developing a B cell epitope-based vaccine is technical; as mentioned many relevant B cell epitopes can be conformational and cannot be isolated from their protein context. However, in the literature there are examples of linear B cell epitopes (synthetic peptides) that induce neutralizing antibodies (e.g. Ramanathan et al., 2016, doi:10.1371/journal.pone.0155900). We can argue that vaccines consisting of entire pathogens induce broader and more lasting immunity than those based on simple pathogen subunits such as antigens. However, vaccines consisting of subunits or coding for subunits can be developed much faster and more safely, which is key in an emergency situation. In fact, subunit/antigen vaccines are becoming dominant, as technological and immunological knowledge increase. In this context, epitope-based vaccines shall represent the next step in the evolution of vaccines since they can focus the immune response towards epitopes of interest. Thereby, B cell epitope prediction is of interest for vaccine design. Since our manuscript is not about epitope-based vaccines we did not include these comments.
Ramanathan, B.; Poh, C.L.; Kirk, K.; McBride, W.J.H.; Aaskov, J.; Grollo, L. Synthetic B-Cell Epitopes Eliciting Cross-Neutralizing Antibodies: Strategies for Future Dengue Vaccine. PLoS One 2016, 11, e0155900–e0155900, doi:10.1371/journal.pone.0155900.
Point 3. Reviewer #2 wonders about the rational for choosing epitopes between 10 and 26 residues
“Minor question: What is the rational for working with epitopes between 10 and 26 residues?”
We chose epitopes between 10 and 26 residues because 98% of the experimental B cell epitopes from antigen-antibody structures had that size. This is now indicated in the revised version of the manuscript, under Material and Methods (page 7, paragraph 1, lines 272-273, highlighted in green).
Point 4. Reviewer #2 suggests that we should mention T cell responses to COVID-19 vaccines
“Issue with the result section and discussion on SARS-Cov-2: the success of the mRNA approach is most likely due to the induction of MHC-I driven CD8 cytotoxic T cell activation, capable of killing infected cells, without ignoring the possible role for neutralizing Abs that would be the result of a process of protein translation and subsequent antigen presentation and MHC-II involvement. However, taking into account the route of administration of the currently approved vaccines, there is no reason to ignore the MHC-I involvement and there is no reason to attribute all the function to antibodies (in that case, an inactivated virus should work as well as a mRNA virus, and there is an agreement that this is not the case – for simple basic immunological reasons). Even in B cell paper, the authors should be more ‘objective/inclusive/neutral’ about this.”
We concur with referee #2 that vaccines based on the spike protein surely induce MHC-I driven Cytotoxic T cells that ought to contribute to the protection provided by mRNA spike vaccines. This is now mentioned in the revised version of the manuscript (page 9-10, section 3.4, lines 345-347, highlighted in green). We have also introduced three new references supporting the new text, which are also highlighted in green in the revised version of the manuscript (refs: 47-49; PMID: 34452047, PMID: 34230917, PMID: 34320609). We realize that the text added is brief but we will hope is enough, as a more extensive commentary on this matter may distort the goal of the work. In fact, we predicted B cell epitopes in SARS-CoV-2 Spike protein just to illustrate the features of BCEPS that help to select predicted B cell epitopes that may induce antibodies recognizing the native antigen for they are accessible, located in loop regions of SARS-CoV-2 spike protein and free of glycosylation.
Point 5. Reviewer #2 poses questions related to SARS-CoV-2 prediction results and B cell epitope vaccines
“Also in the context of SARS-Cov-2: the finding that some of the predicted linear epitopes are confirmed targets of neutralizing antibodies (also mentioned in the discussion), does not prove that these antibodies are the once that do the major job in vivo. It could well be that there is a whole range of conformational epitopes that are also targeted by neutralizing antibodies that have a much higher functionality. How can the authors exclude this? This should be part of the discussion. The overall ‘feeling’ after the review of this paper is that ‘Yes’ it works in one direction: if you have good neutralizing antibodies and you know the sequence and structure of the antigen, you can confirm/predict with the web tool the presence of a linear epitope. But where the article falls short (and to this reviewer this should be a serious open-minded part of the discussion) is: if one takes an antigen and predicts the linear epitope, is there any proof to shows that such epitope(s) can be used in a vaccine approach delivering the efficacy of the full protein? The theoretic discussion on the T cell epitope at the end of the discussion does not answer that question, and without that open discussion, there is a very big question mark as to how useful this approach (web tool) is for the design of the SARS-Cov-3 vaccine we are going to be needing 5 years from now. To this reviewer, this discussion needs to be a big part of the paper.”
The questions posed by the reviewer #2 are indeed interesting and we are happy to comment them. First, we wish to stress that we did not aim to design an epitope-based vaccine against SARS-CoV-2; just to illustrate the use and features of BCEPS. BCEPS is a tool for predicting linear/continuous B cell epitopes and thereby we only checked if BCEPS could identify linear B cell epitopes in SARS-CoV-2 S protein that have been reported to be targeted by neutralizing antibodies. However, as the reviewer suggests, there are also many other conformational epitopes that may be targeted by neutralizing antibodies. In fact, the IEDB database currently includes ~150 distinct conformational B cell epitopes in SARS-CoV-2 (≥ 10 residues) that has been reported to be targeted by neutralizing antibodies. Whether these conformational epitopes are of greater functionally/relevance than linear B cell epitopes is an open question that will require experimental scrutiny. However, unlike linear B cell epitopes, conformational B cell epitopes in SARS-CoV-2 can not be used to produce antibodies unless they are engineered in adequate scaffold simulating their 3D-conformation. Since we do not want to make false claims nor lead to confusion, in the discussion of the revised version of the manuscript we now indicate the following (page 12, lines 477-485):
“It is worth noting that SARS-CoV-2 Spike protein also exhibits many other neutralizing B cell epitopes that are conformational (about 150 of them with ≥ 10 residues can be identified in the IEDB resource) that are out of the reach of BCEPS. Whether these conformational epitopes are of greater functionally/relevance than the linear B cell epitopes is an open question that will require experimental scrutiny. However, unlike linear B cell epitopes, these conformational B cell epitopes in SARS-CoV-2 Spike protein cannot be used to produce antibodies alone, unless they are engineered in adequate scaffolds retaining the epitope 3D-structure.”
With regard to whether there is any proof to show that B cell epitopes predicted by BCEPS can be used in a vaccine approach delivering the efficacy of the full protein, we again do not want to make any type of overstatement. Epitope prediction is routinely used by biotech companies to select peptides that are then used to produce and deliver antibodies for customers, which then test if they work in different experiments (e.g Western Blots, Immunoprecipitation, etc) for their ability to recognize the corresponding antigen. In this context, we showed that BCEPS could identify B cell epitopes that could be relevant to produce antibodies cross-reactive with the native antigens. However, BCEPS, nor any B cell epitope prediction method, can bypass the need for experimental verification. Therefore, B cell epitopes predicted by BCEPS within any given query antigen will need to be synthesized and antibodies produced against them in animal models will need to be tested for their ability to recognize the antigen. These B cell epitopes may be useful for epitope vaccine design only if the raised antibodies have some relevant biological activity against the corresponding pathogen, e.g. blocking viral entry. After the comment of the reviewer, we found convenient to include this text in the revised version of manuscript, under Discussion (page 12, lines 485-491, highlighted in green).
The discussion we had on the ability of BCEPS to predict the presentation of B cell epitopes by MHC II molecules was meant to explain its connection with B cell epitope immunogenicity rather than to address whether B cell epitopes predicted by BCEPS will make a better vaccine than that consisting on the entire antigen and will have an impact on a new SARS-CoV. To make any claim on that sense will be pretentious from our behalf and falls out of the scope of our work. Moreover, we actually believe that the technology is not ready yet for generalized B cell epitope-based vaccines, particularly because many relevant B cell epitopes are conformational. However, technological leaps do occur and what now seems only a possibility could become soon a reality. In that case, tools like BCEPS will help with the initial selection of peptides.
Reviewer 3 Report
The authors have generated a new web tool to be able to select immunnorreactive and cross-reactive linear B cell epitopes. The predictive model was generated with a limited number dataset. However, the predictive capacity of the web tool seems robust.
That been said, there are some minor issues to be addressed befor the manuscript is suitable for publication.
In section 2.4 the number (3) should be aligned with the others (in the algorithm section.
The authors shoul improve the quality of the figures. Figure 1 is a bit pixeled, and Figure 2b is blurry.
Limited number we constructed a non-redundant dataset including the sequences of 555 B cell epitopes and 555 non-B cell epitopes
Author Response
We wish to thank reviewer #3 for the time and work reviewing our manuscript, and for his/her positive consideration and encouraging comments. Reviewer #3 has only some minor comments, which we address next.
Point 1. Reviewer #3 has noted a formatting error
“In section 2.4 the number (3) should be aligned with the others (in the algorithm section)”
We appreciate the observation of the reviewer. In the revised version of the manuscript the equation numbers are all aligned. We hope they stay that way.
Point 2. Reviewer #3 suggests improving the quality of the figures
“The authors should improve the quality of the figures. Figure 1 is a bit pixeled, and Figure 2b is blurry.”
We provided a set of high-definition figures but those embedded in the manuscript did not have such resolution and/or may have lost resolution after uploading the manuscript. In any case, in the revised version of the manuscript we have now inserted higher resolution figures.
Point 3. Reviewer #3 observes that the number of epitopes used for training is limited.
“Limited number we constructed a non-redundant dataset including the sequences of 555 B cell epitopes and 555 non-B cell epitopes”
This is a fine appreciation indeed. The B cell epitope sequence space is certainly much larger than our dataset and it is wise to wonder if we have enough epitopes to model the B cell epitope space. We cannot answer this question with certainty. In fact, it is an open question how much data is required for Machine Learning, as more data does not necessarily lead to better models. However, there is no doubt that the quality and diversity of the data is of paramount relevance for Machine Learning. Therefore, we used B cell epitope sequences obtained from the 3D-structure of antibody-antigen complexes and applied methods to reduce sequence similarity in our datasets. As a result, we produced B cell prediction models that showed a good generalization power, outperforming competing methods in independent datasets. Said so, we should also add that the number B cell epitopes we used for training is in the same order of magnitude than that used in other methods.
Round 2
Reviewer 2 Report
Thank you for addressing the concerns flagged in the 1st round of reviewing.
Author Response
Following the reviewer’s comments, in the revised version of the manuscript, we have now included some new text under Discussion (page 12, lines 491-494, highlighted in red) indicating the following:
“Currently, there is no commercial vaccine available based on B cell epitope predictions. Moreover, depending on the infection/pathogen, inducing antibodies with the desired specificity can be an important step towards developing a vaccine, but antibodies alone may not suffice”.
We hope we have now properly addressed the reviewer's concerns and thank him/her for reviewing our manuscript.